# Assessment of Mental Health Comorbidities and Relief Factors in Moroccan Women during the Third Trimester of Pregnancy: A Cross-Sectional Study

**DOI:** 10.3390/healthcare12151470

**Published:** 2024-07-24

**Authors:** Maroua Guerroumi, Amina Aquil, Noura Dahbi, Ouassil El Kherchi, Salma Ait Bouighoulidne, Soumia Ait Ami, Meryam Belhaj Haddou, Arumugam R. Jayakumar, Abdeljalil Elgot

**Affiliations:** 1Epidemiology and Biomedical Unit, Laboratory of Sciences and Health Technologies, Higher Institute of Health Sciences, Hassan First University of Settat, Settat 26000, Morocco; m.guerroumi@uhp.ac.ma (M.G.); amina.aquil@uhp.ac.ma (A.A.); no.dahbi@uhp.ac.ma (N.D.); e.ouassil@uhp.ac.ma (O.E.K.); s.aitbouighoulidne@uhp.ac.ma (S.A.B.); s.aitami.res@uca.ac.ma (S.A.A.); 2University Teaching Hospital Mohammed VI, Marrakech 40070, Morocco; m.belhajhaddou@uhp.ac.ma; 3Department of Obstetrics, Gynecology & Reproductive Sciences, Miller School of Medicine, University of Miami, Miami, FL 33136, USA; ajayakumar@med.miami.edu

**Keywords:** mental comorbidities, social support, third trimester of pregnancy

## Abstract

Background: During pregnancy, women can experience mental alterations, particularly anxiety and depression, which mark an important transition period in their lives. Social support appears to be a crucial alleviating factor for these disorders. The aim of this study is to assess the extent of psychological disturbances and their relieving factors by investigating correlations between mental status and different sociodemographic and clinical characteristics during the third trimester of pregnancy. Methods: A cross-sectional study including 160 pregnant women in their last trimester was carried out in Morocco, notably at the Ibn Sina University Hospital and in two health centers. A pre-structured questionnaire, including sociodemographic and clinical variables and internationally recognized scales such as the Multidimensional Scale of Perceived Social Support (MSPSS), the Perceived Stress Scale (PSS), the Pittsburgh Sleep Quality Index (PSQI), the Epworth Sleepiness Scale (ESS), the Bergen Insomnia Scale (BIS), and the Hospital Anxiety and Depression Scale (HADS), was mobilized. Results: The prevalence of depression and anxiety was 18.75% and 12.5%, respectively. A correlation between these two mental disorders and the level of education, pregnancy planning, monthly income, and provision of health coverage was found (*p*-value < 0.05). The main determinants of anxiety were stress (*p*-value = 0.047) and social support (*p*-value < 0.001), while depression was limited to social support (*p*-value < 0.001) and sleep quality (*p*-value = 0.015). Conclusions: It is essential to take action against these disorders and their predictive factors by raising awareness and implementing a diagnosis and care protocol with healthcare professionals to guide and orient distressed women.

## 1. Introduction

From conception to the baby’s first year of life, the perinatal period is a major psychological and social transition [1]. It indicates that there is no protection from the emergence of various mental disorders during pregnancy, whether they arise from the decompensation of a prior illness or initially occur during the gestational period [2]. According to Andersson et al. (2003), psychiatric disorders were present in 14.1% of women, of whom 3.3% suffered from major depression and 6.9% from minor depression, while 6.6% were anxious and only 5.5% received therapeutic treatment [3]. Similarly, American studies suggest that less than a third of the women suffering from psychiatric disorders during pregnancy have access to appropriate treatment [4].

Nevertheless, depression is one of the most common psychiatric and mood disorders. It is not uncommon during pregnancy. The WHO reports that worldwide, around 10% of pregnant women and 13% of women who have just given birth suffer from mental disorders, mainly depression [5]. As stated by Lee and his colleagues (2007), anxiety seems to be highly prevalent during pregnancy and is the most common psychological disorder in the perinatal period. Anxiety levels tend to be higher in the first and last trimesters [6]. Pregnant women who experience depression, however, are less likely to seek prenatal care [7]. In addition, they have a higher chance of experiencing issues during pregnancy and labor and a higher risk of suicide [8]. Pregnancy-related anemia, heightened medical risk, prenatal hypertension syndrome (pre-eclampsia), and cesarean birth are among the issues linked to anxiety throughout pregnancy [9]. Postpartum mental discomfort, including anxiety and depression, is more likely to occur in women who have anxiety disorders during pregnancy [10].

These mental comorbidities encountered at this precious moment may be associated with a variety of factors, including socio-demographic characteristics, emotional stability, relationships with husband and mother, cultural attitudes, preparation for motherhood, previous psychological problems, the presence or absence of a child, and the existence or absence of obstetrical complications during pregnancy [11].

During pregnancy, sleep quality declines, with notable variations from one trimester to the next. Certain sleep problems, in particular insomnia, sleepiness, sleep-disordered breathing, and restless legs syndrome, may become more frequent as a result [12,13,14]. Data from the literature show that sleep disorders can be predictive of depressive symptoms and that poor sleep quality, particularly in the second and third trimesters of pregnancy, is directly linked to depressive symptoms at the end of pregnancy [15,16]. In addition, a study of pregnant women in Finland showed that mild to moderate levels of depressive and anxiety symptoms were associated with sleep disturbance in late pregnancy [17].

Furthermore, studies have confirmed the relief effect of social support, showing that women whose level of social support from husbands, family, and friends is inadequate have difficulty adapting to pregnancy and suffer from psychosocial problems [18,19]. The presence of support systems reinforces self-control and positive emotions and makes certain pregnancy-related changes less stressful for the pregnant woman [11,19]. However, intense stress in pregnant women can be toxic due to the release of high cortisol levels. This can have negative repercussions on the mother’s health, the progress of the pregnancy, and the health of the child [20].

In reviewing the literature and previous research on the psychosocial aspects of pregnancy, we found that few works and empirical data have addressed the question of the relationship between psychological problems and other clinical and sociodemographic factors, particularly social support, especially in the Moroccan context. This finding was instrumental in planning our study, which aimed to assess the extent of mental disorders, particularly anxiety and depression, and their predisposing factors by investigating correlations between mental status and various demographic, clinical, and behavioral characteristics during the third trimester of pregnancy.

## 2. Material and Methods

### 2.1. Type and Setting of Study

This is a cross-sectional, longitudinal study that was conducted in the Moroccan capital city “Rabat”, specifically in two urban health centers, i.e., the first-level Ocean and Diour Jmaa and the maternity unit of the IBN Sina University Hospital. The study took place between 12 May 2021 and 10 June 2022.

### 2.2. Participants

The target population for this study was pregnant women in the 3rd trimester of pregnancy. The sample was constructed randomly using a non-probability sampling method, with 210 pregnant women at the outset and only 160 pregnant women meeting the inclusion criteria.

(a) Inclusion criteria:

(1) Women in their final trimester of pregnancy; (2) normal pregnancy progression with no history of hospitalization; (3) capable of comprehending, expressing, and speaking Moroccan dialectal Arabic; (4) well-attended pregnancy since the first trimester; and (5) informed consent was obtained from all individual participants included in this study.

(b) Exclusion criteria:(1) Women with gestational pathologies (gestational hypertension and diabetes, anemia, thromboembolic disorders, etc.); (2) women with pathologies prior to pregnancy; (3) women with multiple pregnancies; (4) foreign pregnant women; and (5) women with a history of psychiatric disorders.The data were collected in a confidential environment with the help of health professionals working in the health facilities. Participants were randomly selected from among those receiving medical follow-up. Next, the principal investigator begins to present the purpose of the study to the participants, then explains each question and ticks the answers chosen by the woman using pre-structured questionnaires. This procedure took about 25 min to complete.

### 2.3. Measures

#### 2.3.1. Sociodemographic Data

It includes a range of questions enabling us to get closer to the participant and identify certain characteristics relating to marital status, age, level of education of the wife and husband, profession, economic status, the presence of basic medical coverage, in particular the Medical Assistance Plan (RAMED), the National Fund for Social Welfare Organizations (CNOPS), the National Social Security Fund (CNSS), or other insurance and data relating to the current and previous pregnancy (parity, gestational age, monitoring, and progress of the pregnancy). Finally, the women were also questioned about the existence of psychiatric antecedents (circumstances of onset, nature, and treatment established).

#### 2.3.2. Multidimensional Scale of Perceived Social Support (MSPSS)

It provides a subjective assessment of the perceived availability of social support across three specific groups, i.e., husband, family, and friends or significant others. Empirical evidence also shows that this scale has good internal reliability and moderate construct validity. It comprises 12 items; each item is scored from 1 to 7, “very strongly disagree to very strongly agree”. The total score ranges from 12 to 84, and a high score (over 61) indicates a high perception of social support [21]. In this study, Cronbach’s alpha coefficient is 0.80, indicating good reliability.

#### 2.3.3. Perceived Stress Scale (PSS)

Developed in English, it was then translated and validated in Arabic. Researchers then adapted it to the psychometric properties of pregnancy and postpartum. The scale consists of 10 items, each of which indicates the extent to which pregnant women in the third trimester of pregnancy, particularly in the last month, anticipate that their current life situations are perceived as stressful, distressing, threatening, and uncontrollable, using a 5-point Likert scale (0 = never, 1 = almost never, 2 = sometimes, 3 = quite often, and 4 = very often). The total score ranges from 0 to 40, with higher scores indicating greater perceived stress [22,23]. Cronbach’s alpha for the PSS is 0.78, which shows good validity.

#### 2.3.4. Pittsburgh Sleep Quality Index (PSQI)

Questionnaire designed to self-assess sleep quality over the previous month, using several criteria, i.e., sleep habits, subjective sleep quality, sleep latency, sleep duration, habitual sleep efficiency, sleep disorders, use of sleep medication, and also poor daytime form. It contains 19 questions associated with the 5 questions the spouse or roommate were asked. To calculate the score, only the 19 questions are taken into account; these questions combine to give 7 “components”, with the overall score (the sum of the components) ranging from 0 to 21 points, with 0 meaning there are no difficulties and 21 indicating the presence of major difficulties. Usually, an overall score higher than 5 is an indicator of sleep disorders, and this scale is developed in English, then translated and validated in Arabic [24,25]. Cronbach’s alpha for this work is 0.82, which indicates the good reliability of the used questionnaire.

#### 2.3.5. Epworth Sleepiness Scale (ESS)

Developed by Dr. Murray Johns of the Epworth Hospital in Melbourne, Australia, in 1991, it provides a subjective assessment of the propensity to fall asleep during the day in everyday situations of relative inactivity. It comprises 8 questions, which are scored from 0 to 3. The overall score corresponds to the sum of 8 questions ranging from 0 to 24, i.e., below 11: normal vigilance; from 11 to 16: excessive daytime sleepiness; and above 16: severe daytime sleepiness [26]. Cronbach’s alpha in this study is 0.873, which demonstrates good internal validity.

#### 2.3.6. The Bergen Insomnia Scale (BIS)

This questionnaire comprises six items, the first four of which provide information on nocturnal conditions such as a delay of more than 30 min in falling asleep, waking up more than 30 min during the night, waking up more than 30 min earlier than expected, and not feeling sufficiently rested after sleep. These items correspond to DSM-IV-TR criterion A for insomnia. The last two items assess daytime functioning due to sleepiness and sleep inefficiency, which corresponds to DSM-IV-TR criterion B. Women are diagnosed as insomniacs if they have had at least one A and one B criterion on three or more days per week over the past month. The BIS represents validity against polysomnographic data and some self-report scales [27,28]. The value of Cronbach’s alpha for the BIS was 0.73.

#### 2.3.7. The Hospital Anxiety and Depression Scale (HADS)

Developed by Zigmond and Snaith in 1983, it aims to identify the presence of anxiety–depression symptomatology and assess its severity. The questions are designed to avoid somatic aspects, which can alter the assessment. It comprises 7 items to assess depression and 7 items to assess anxiety. The sum of each score is used to identify patients with depressive or anxious symptoms. The scores are as follows: from 0 to 7: no disorders; from 8 to 10: suspected disorders; from 11 to 21: confirmed disorders; and the combination of the two indicates the presence or absence of an anxiety–depression syndrome (translated into Arabic version) [29,30].

This questionnaire has been adapted to our Moroccan context and to our usable Arabic dialect by the intervention of a number of experts specializing in the process of translation and retranslation in order to collect reliable and credible data.

### 2.4. Data Processing

The data collected using the questionnaire were processed using SPSS version 22 software. Univariate and bivariate descriptive statistics were performed. To test the correlation between two quantitative variables, Pearson’s or Spearman’s correlation tests were performed. Similarly, to test the correlation between a quantitative variable and a two-modality qualitative variable, the Student’s *t*-test for two independent samples or the Mann–Whitney test were also useful, depending on whether or not the normality hypothesis and homogeneity were verified. To test the relationship between a quantitative variable and another qualitative variable with more than two modalities, the ANOVA, or Kruskal–Wallis test, was used. To identify the factors determining anxiety and depression, multiple linear regression was used. Associations with a *p*-value ≤ 0.05 are considered significant.

## 3. Results

### 3.1. Description of the Survey Population

This study included 160 women who were pregnant in their last trimester. More than half (66.88%) of these women were aged between 25 and 35; most of the women were married; 76.2% complained of a monthly income of less than DH 4000 (DH 4000 ≈ USD 400); only 22.5% of patients had access to higher education; and around 5.6% of patients lived in rural areas. With regard to the benefits of the healthcare system, 159 women had health coverage under RAMED, CNOPS, CNSS, or others; 61.3% of pregnancies were planned; and almost half of the women surveyed had no previous experience of pregnancy. In addition, we found that 18.75% of the pregnant women in our sample complained of depression during the last trimester of pregnancy, and only 12.50% suffered from anxiety (Table 1).

### 3.2. Association between Anxiety, Depression, and Socio-Demographic and Clinical Characteristics

In this cross-sectional study, the development of anxiety in the third trimester of pregnancy was related to various sociodemographic and clinical variables, including level of education, economic status, planned pregnancy, and the presence of health coverage, with a significance level of less than 0.01, whereas other variables were not statistically associated with this type of mental disorder (Table 2).

With regard to depression, the results obtained in the present study revealed a strong correlation between this type of mental disorder and economic status, level of education, housing, and the number of births experienced, as well as pregnancy planning and the existence of medical coverage (*p*-value < 0.05) (Table 3).

In fact, these findings show that advancing age, a precarious living situation, a lack of health insurance, and an unplanned pregnancy contribute to the development of mental distress in the last trimester, while for depression, the effects of parity and housing are added.

### 3.3. Predictive and Relieving Factors for Comorbidities in Mental Health

The use of a multiple linear regression model in the present study revealed an association between sleep quality and depression, with a significance level of 0.015. In addition, pregnant women who complained of poor sleep quality during the third trimester were more likely to suffer from this type of mental disorder. Furthermore, pregnant women who had experienced stressful events during this period of gestation were more likely to present signs of anxiety alone, with a *p*-value of 0.47. While the social support received from the husband, family, or friends proved to be a relieving factor against the development of these mental disorders (*p* value < 0.001), this result shows the positive effect of psychosocial support and accompaniment during this critical phase, as it enables women to adapt easily to this transition to motherhood away from mental comorbidities, whereas no association was found between anxiety and depression and the two sleep disorders, mainly insomnia and somnolence, their effect being negligible (Table 4 and Table 5).

## 4. Discussion

Given the paucity of research and data on the description and analysis of changes in the mental state of Moroccan pregnant women during the last trimester, we carried out a study in this context in order to dissect and approach this aspect. The study was conducted in the Moroccan capital among women in their third trimester of pregnancy. Around half of the women recruited had no previous experience with pregnancy, which obviously explains the anxiety, fear, stress, and mood swings they experience when faced with the physiological, hormonal, and psychological changes associated with gestation.

Pregnancy, particularly the last trimester, is a period of emotional and psychological upheaval that can lead to an increase in mental disorders, particularly depression and anxiety. The results obtained in our study show that prenatal depression is found in 18.75% of pregnant women, and only 12.50% suffer from signs of anxiety. These two prevalences are very high compared with those reported in a prospective study conducted among Dutch women at the beginning and end of pregnancy, evaluating the influence of maternal characteristics on these mental disorders at the time of gestation [31]. This difference in percentage can be explained by the methodology used and the data collection tool employed, as well as the gestational period evaluated. Indeed, according to Verreault et al. (2014), there is evidence that these two psychological disturbances are also highly comorbid during the prenatal period and generate the anxiety–depression syndrome, although they probably share most of the well-established psychosocial risk factors of depression or anxiety alone [32].

During this work, it emerged that a deteriorating economic situation exists. Approximately 76.2% of our populace earns less than DH 4000 (DH 4000 = USD 400) per month. Even still, there is a clear correlation between the emergence of these mental illnesses and families’ financial hardships. Meanwhile, a study of Italian pregnant women during the last trimester showed, using a univariate analysis method, significant differences between women suffering from an anxiety–depression syndrome (comorbidity) and those complaining of a single morbidity with regard to economic status [33]. Furthermore, level of education may also play a role in the genesis of these two psychological disorders, which was observed when the data were processed with a high significance threshold, unlike age. In the same context, a study carried out in 2013 reported that demographic variables such as age, education, and profession were not associated with prenatal depression [34]. In contrast, Vandelo et al. (2018) report that women aged ≥35 years were more likely to suffer from probable depression, and a higher percentage of women aged ≤30 years complained of probable anxiety. They also seem to suggest that a low educational level increases depressive and even anxious symptoms at the beginning and end of pregnancy [31].

As far as the number of births or parity and housing are concerned, these two parameters are strongly associated with depression, which means that survival in precarious housing as well as the experience of giving birth, whether for the first time or in the face of a traumatic or violent history, make women more vulnerable to depressive symptoms only during the third trimester of pregnancy. In perfect agreement, a meta-analysis of a precise number of studies published in indexed journals predicts that parity is one of the preliminary factors for the onset of prenatal depression [35].

What is more, the presence of health insurance provides women with comfort during this critical period, enabling them to access the healthcare system more easily and benefit from certain services offered by the various health facilities. This factor reduces the likelihood of suffering from prenatal anxiety and depression. Moreover, 72.5% of our samples were covered by health insurance. On the other hand, a study carried out by Yohannes Dibaba and his colleagues (2013) in South-West Ethiopia shows that women declaring an unwanted pregnancy are almost twice as likely to be depressed as women whose pregnancy is planned [36]. Similarly, a study defining the etiologies of anxiety at the time of gestation revealed a significant correlation between anxiety and pregnancy planning [37]. The current study sustained these associations.

During the last trimester, women were likely to experience certain sleep disorders that affected sleep architecture, efficiency, and overall quality. We found that 81.87% of our sample complained about sleep disturbances. In fact, this magnitude can be linked to a number of factors, including increased uterine volume, pollakiuria, ligament pain, uterine contractions, gastro-esophageal reflux, and fetal movements [38]. Based on a multiple linear regression model, we found that poor sleep quality is a strong predictor of depression. This result is in line with a study carried out in Portugal on 143 women in the same physiological situation, which showed that pregnant women suffering from depression had a greater number of awakenings during the night and spent more hours trying to fall asleep compared with non-depressed pregnant women [39]. Within the same framework, research indicates that sleep disturbances in late pregnancy are inversely correlated with anxiety and depression [17]. Additionally, physical symptoms and poor sleep quality during the first trimester were linked to depressive symptoms at the end of pregnancy, according to a 2010 study by Kamysheva and colleagues on 257 healthy pregnant women [40].

Despite the fact that insomnia and sleepiness represent major cross-over difficulties at the time of gestation, in our sample, we found that 86% and 73.75%, respectively, suffered from these sleep disorders, and no association with the aforementioned mental disorders was proven. Conversely, Linda Aukia et al. (2020) pointed out that women who suffered more from depressive and anxiety symptoms had higher levels of insomnia and sleepiness at each stage of pregnancy, with a representative significance threshold [41]. However, a cross-sectional study of pregnant women in Ghana showed that insomnia has been found to be a significant risk factor for prenatal anxiety [42].

In other words, women at this pivotal time need the support of their husbands, family, and friends to overcome these psychological imbalances, particularly anxiety and antenatal depression. This finding was observed in our studies, half of which perceived an adequate level of support. In line with a review of the literature highlighting the important role of social support during the perinatal phase against depression and anxiety [43], these outcomes explain the positive effect of psychosocial support on the mental state of pregnant women at the end of pregnancy through the satisfaction of emotional needs and expectations.

In fact, the analysis of the interrelationship between stress, anxiety, and depression proves that confrontation of stressful events contributes to the onset of anxiety during the third trimester and not depression, with a *p*-value of less than 0.05, whereas 91.87% of the women surveyed had a moderate level of stress. At the same time, a prospective cohort showed that perceived stress was an important predictor of comorbid anxiety and mild-to-severe and moderate-to-severe depressive symptoms [44]. Similarly, a longitudinal study found that women with negative life experiences reported more mental disturbances throughout pregnancy. Indeed, this confrontation increased the risk of depressive symptoms by 0.7 and that of anxiety signs by 1.8 at the end of gestation [31].

This study has certain limitations, notably the fact that the data were collected by self-reporting and that the sample may not be representative of the whole of Morocco, as well as the fact that it focuses only on the third trimester. In addition, future research will extend to the nine months of pregnancy and even to the postpartum period in order to explore this area in greater depth. Furthermore, some of the scales used have not undergone psychometric validation. However, its strength lies in the fact that it deals with an aspect that is ignored in the Moroccan context.

## 5. Conclusions

The last trimester of pregnancy represents a period of heightened vulnerability during which women are predisposed to a number of physical and psychological problems and discomforts, in particular the development of anxiety and depression. These disorders can be associated with a range of socio-demographic and clinical factors, as well as the experience of stressful events and poor sleep quality. On the other hand, social support appears to be a factor in alleviating these mental disorders. The latter can be detrimental to maternal and fetal health in the short and long term. Awareness-raising, early diagnosis, and optimal care must be put in place.

## Figures and Tables

**Table 1 healthcare-12-01470-t001:** Description of the target population (n = 160).

Variable		Frequency (%)
Age (years)	<25	43 (26.87)
	25–35	107 (66.88)
	>35	10 (6.25)
Marital status	Married	157 (98.1)
	Remarried	1 (0.6)
	Single	2 (1.3)
Education level	None	34 (21.3)
	Primary	30 (18.8)
	College	14 (8.8)
	Secondary education	46 (28.8)
	Higher education	36 (22.5)
Habitat	Urban	151 (94.4)
	Rural	9 (5.6)
Economic situation (DH 10 ≈ USD 1)	<DH 2000	25 (15.6)
	DH 2000–4000	97 (60.6)
	DH 4000–10,000	38 (23.8)
Parity	Primiparous	69 (43.1)
	Multiparous	91 (56.9)
Health coverage	None	44 (27.5)
	CNSS	47 (29.4)
	CNOPS	26 (16.3)
	RAMED	42 (26.3)
	Other	1 (0.6)
Planned pregnancy	Yes	98 (61.3)
	No	62 (38.8)
Anxiety	Yes	20 (12.5)
	No	140(87.5)
No	Yes	30 (18.75)
	No	130 (81.25)

**Table 2 healthcare-12-01470-t002:** Association between HADS-A and socio-economic variables. HADS-A: Hospital Anxiety and Depression Scale-Anxiety.

Variable		Mean (sd), Range	*p*-Values
Age (years)	≤25	7.02 (2.01), 6.46–7.57	0.192
	25–35	7.16 (2.22), 6.72–7.62	
	>35	8.5 (2.83), 6.47–10.53	
Level of education	None	7.97 (2.41), 7.13–8.81	<0.001
	Primary	7.70 (2.23), 6.87–8.53	
	College	8.21 (1.05), 7.61–8.82	
	Secondary	7.22 (1.95), 6.64–7.8	
	Higher education	5.64 (1.88) 5–6.28	
The habitat	Urban	7.13 (2.22), 6.77–7.49	0.054
	Rural	8.63 (1.40), 7.45–9.80	
Economic situation (DH 10 ≈ USD 1)	<DH 2000	7.4 (2.59), 6.33–8.47	0.008
	DH 2000–4000	7.75 (1.92), 7.36–8.14	
	DH 4000–10,000	5.69 (1.96), 5.03–6.36	
	>DH 10,000	5 (1.41), 0–17	
Parity	Primiparous	6.87 (2.13), 6.36–7.38	0.135
	Multiparous	7.45 (2.24), 6.98–7.92	
Planned pregnancy	Yes	6.71 (2.11), 6.29–7.14	0.001
	No	7.97 (2.15), 7.42–8.52	
Health coverage	Yes	6.91(2.30), 6.49–7.34	0.004
	No	7.93 (1.78), 7.40–8.47	

**Table 3 healthcare-12-01470-t003:** Association between HADS-D and socio-economic variables. HADS-D: Hospital Anxiety and Depression Scale-Depression.

*Variable*		Mean (sd), Range	*p*-Values
Age (years)	≤25	7.24 (2.65), 6.52–7.97	0.32
	25–35	7.66 (2.46), 7.16–8.16	
	>35	8.40 (2.22), 6.81–9.99)	
Level of education	None	8.68 (1.91), 8.01–9.35	<0.001
	Primary	8.30 (1.76), 7.64–8.96	
	College	8.86 (2.62), 7.34–10.37	
	Secondary	7.17 (2.41), 6.46–7.89	
	Higher education	5.89 (2.69), 4.98–6.80	
The habitat	Urban	7.45 (2.52), 7.04–7.87	0.005
	Rural	9.75 (1.03), 8.88–10.62	
Economic situation (DH 10 ≈ USD 1)	<DH 2000	8.44 (2), 7.61–9.27	<0.001
	DH 2000–4000	8.10 (2.2), 7.66–8.55	
	DH 4000–10,000	5.53 (2.49), 4.69–6.37	
	>DH 10,000	7 (5.65), 0–57	
Parity	Primiparous	7.06 (2.58), 6.44–7.68	0.023
	Multiparous	7.95 (2.40), 7.44–8.45	
Planned pregnancy	Yes	6.92 (2.41), 6.44–7.40	<0.001
	No	8.58 (2.35), 7.98–9.18	
Health coverage	Yes	7.27 (2.56), 6.79–7.74	0.024
	No	8.31 (2.23), 7.64–8.98	

**Table 4 healthcare-12-01470-t004:** Predictive factors of anxiety (HADS-A) in our target population.

	B	t	*p*-Value	CI of 95%	
Constant		3.647	0.000	4.065	13.672
Perceived Stress	0.142	2.003	0.047	0.001	0.220
Social Support	−0.593	−8.869	0.000	−0.094	−0.060
Sleep Quality	0.085	1.309	0.193	−0.032	0.160
Insomnia	0.075	1.208	0.229	−0.023	0.097
Sleepiness	−0.036	−0.605	0.546	−0.099	0.052

B: Standardized coefficients of regression. *p*-value: Level of significance. Confidence interval: CI.

**Table 5 healthcare-12-01470-t005:** Predictive factors of anxiety (HADS-D) in our target population.

	B	t	*p*-Value	CI of 95%	
Constant		4.474	0.000	7.725	19.942
Stress	0.022	0.277	0.782	−0.119	0.158
Social Support	−0.535	−7.166	0.000	−0.101	−0.057
Sleep Quality	0.178	2.453	0.015	0.030	0.274
Insomnia	−0.086	−1.233	0.220	−0.124	0.029
Sleepiness	−0.033	−0.496	0.621	−0.120	0.072

B: Standardized coefficients of regression. *p*-value: Level of significance. Confidence interval: CI.

## Data Availability

The corresponding author can provide the datasets utilized and/or analyzed in this study upon formal request.

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
