# Peer review of "Assessment of Mental Health Comorbidities and Relief Factors in Moroccan Women during the Third Trimester of Pregnancy: A Cross-Sectional Study"

_healthcare, 2024, doi:10.3390/healthcare12151470_

Round 1

Reviewer 1 Report

Comments and Suggestions for Authors

In this study Guerroumi et al aimed to assess the connection between certain socio-demographics and clinical data, depression and anxiety in pregnant women of the third semester.

Although the purpose seems interesting, the results and discussions do not sustain properly the intense evaluation and aspects that were considered in the study. For example, investigators evaluated sleep with three instruments, but in the results and discussion sections the information is scanty and unclear. The Pittsburgh Sleep Quality Index is very complex and evaluates sleep from many facets which were not clearly stated in this study. I would suggest improving the quality of this study by working especially on results and their value in the discussion. Please consider also, the following observations.

Line 36: “does not protect…” please rephrase.

Line 37: “mental comorbidities” may be replaced with “mental disorders” 

Line 42: “less than a third of the women”, “the” is missing.

Line 49: “and the last trimester”, “the” is missing.

Line 52: two verbs are missing, and the sentences do not make any sense, please correct them.

Line 55: anxiety disorders or depression or “and”, it is not clear …

Line 62: “Sleep quality declines…and these alterations…” as far as I understand you only refer to sleep quality, which are the others?

Lines 95-96: Subjects included cannot meet both inclusion and exclusion criteria, please correct the statement.

Lines 100-101: There is not clear, please rephrase.

Line 112: the verb “take” should be in past tense (took).

Line 169:  You cited DSM-IV here, but without mentioning the pages, I do not understand how this reference was significant here.

Line 218: There is a comma following a full stop, please correct the mistake.

Lines 248-251: The sentence does not make any sense.

Line 268:  The meaning of the phrase “certain pleasure” in this context is not clear.

Comments on the Quality of English Language

English quality required, especially regarding phrasing.

Author Response

Dear reviewer,

Thank you very much for your valuable suggestions and comments on our manuscript. Those comments are of great assistance to me for improving and revising our manuscript. We have studied comments carefully and have made correction in line with the suggestions.

Please find in the attached file our responses to your comments.

Many thanks

Reviewer 2 Report

Comments and Suggestions for Authors

Thank you for the opportunity to read the article Assessment of Mental Health Comorbidities and Relief Factors in Moroccan Women during the Third Trimester of Pregnancy: A Cross Sectional Study

The article focuses on the topic, is well-founded and methodologically justified. It is an exploratory study that uses a set of scales to illustrate the relationship between depression and anxiety during pregnancy. The results of the study are presented clearly as well as the discussion.

I suggest that the authors: 

- present an analysis of the potential and limits of the study;

-  in the conclusion  respond to the objective of the study unequivocally, that is, illustrating which factors interfere negatively and which interfere positively in this specific case.

I wish you good work

Author Response

(The authors gave the same response as above.)

Reviewer 3 Report

Comments and Suggestions for Authors

This cross-sectional study examined correlates of depressive and anxiety symptoms among 160 pregnant women in their last trimester in Morocco. The authors reported that while the major determinants of anxiety were stress and social support (p <0.05), sleep quality and social support were significantly (p <0.05) associated with depression. Though these findings are unsurprising, the African context makes the study unique.

Given the length of the manuscript, I wonder if this can be considered for publication as a research note rather than a full article. Regardless, careful line editing by a native English speaker is strongly recommended.   

Comments on the Quality of English Language

Careful line editing by a native English speaker is strongly recommended.  

Author Response

(The authors gave the same response as above.)

Round 2

Reviewer 1 Report

Comments and Suggestions for Authors

The quality of the manuscript has been significantly improved after revision. There are still some issues to address.

Comments:

Lines 99-107: The inclusion criteria need some additional polish, especially rephrasing to be aligned with each other. For example at inclusion criteria 2, you use present tense, but at criteria 4, you use present perfect. Regarding criteria 5, you mention “written and signed consent” …”was given” could be needed to make sense. At exclusion criteria 1, “gravidic’ may be replaced with “gestational” and at criteria 5, it would sound more clearer “women with a history of …”

Line 218: The verb “obtain” is used in past tense, so should be the verb “reveal”.

Lines 220-221: I do not grasp the meaning of the sentence “as well as planned pregnancy…” they are correlated with economic status, level of education and so on?, or the planed pregnancy is correlated with the medical coverage? Please clarify.

Lines 231: You mention of “a significant lefel of 0.15”, maybe you intended to say “0.015” as in the table, otherwise it is not statistically significant and should be changed.

Line 237: The sentence should end after the p-value and start another one, otherwise the meaning is hard to grasp.

Line 294: The verb “carry out” is spelled wrongly, please correct it.

Lines 298-299: “The current investigation..” I do not understand the meaning of the sentence, please rephrase.

Comments on the Quality of English Language

English quality: could still be improved

Author Response

Dear reviewer,

Thank you very much for your valuable suggestions and comments on our manuscript. Those comments are of great assistance to me for improving and revising our manuscript. We have studied comments carefully and have made correction in line with the suggestions.

Best regards
